# Barrier Protection and Recovery Effects of Gut Commensal Bacteria on Differentiated Intestinal Epithelial Cells In Vitro

**DOI:** 10.3390/nu12082251

**Published:** 2020-07-28

**Authors:** Nooshin Mohebali, Katharina Ekat, Bernd Kreikemeyer, Anne Breitrück

**Affiliations:** Molecular Bacteriology, Institute of Medical Microbiology, Virology and Hygiene, University Medicine Rostock, 18057 Rostock, Germany; nooshin.mohebali@med.uni-rostock.de (N.M.); katharina.ekat@med.uni-rostock.de (K.E.); bernd.kreikemeyer@med.uni-rostock.de (B.K.)

**Keywords:** IBD, Caco-2 cells, HT29-MTX cells, commensal bacteria, Faecalibacterium prausnitzii, Bacteroides faecis, Roseburia intestinalis, transepithelial electrical resistance (TEER), tight junction proteins

## Abstract

Alterations in the gut microbiota composition play a crucial role in the pathogenesis of inflammatory bowel disease (IBD) as specific commensal bacterial species are underrepresented in the microbiota of IBD patients. In this study, we examined the therapeutic potential of three commensal bacterial species, *Faecalibacterium prausnitzii* (*F. prausnitzii*), *Roseburia intestinalis* (*R. intestinalis*) and *Bacteroides faecis* (*B. faecis*) in an in vitro model of intestinal inflammation, by using differentiated Caco-2 and HT29-MTX cells, stimulated with a pro-inflammatory cocktail consisting of interleukin-1β (IL-1β), tumor necrosis factor-α (TNFα), interferon-γ (IFNγ), and lipopolysaccharide (LPS). Results obtained in this work demonstrated that all three bacterial species are able to recover the impairment of the epithelial barrier function induced by the inflammatory stimulus, as determined by an amelioration of the transepithelial electrical resistance (TEER) and the paracellular permeability of the cell monolayer. Moreover, inflammatory stimulus increased claudin-2 expression and decreased occludin expression were improved in the cells treated with commensal bacteria. Furthermore, the commensals were able to counteract the increased release of interleukin-8 (IL-8) and monocyte chemoattractant protein-1 (MCP-1) induced by the inflammatory stimulus. These findings indicated that *F. prausnitzii*, *R. intestinalis* and *B. faecis* improve the epithelial barrier integrity and limit inflammatory responses.

## 1. Introduction

The intestinal mucosa is a critical barrier that separates the inner and outer areas of intestinal lumen and is essential for the maintenance of mucosal homeostasis. This barrier consists of a single epithelial cell layer, an intraepithelial tight junction (TJ) complex and is considered to be the first line of defense against the hostile luminal environment [1]. These junctional complexes comprise TJ proteins such as claudins, occludin, the zonula occludens, junctional adhesion molecule, and the adherence junction protein E-cadherin, located at the apical-lateral membrane [2,3]. These proteins regulate the passage of solute and macromolecules through the paracellular pathway, thereby playing a vital role in regulating intestinal barrier function [1]. In principle, intestinal permeability reflects the state of the epithelial TJ [4]. The dysregulation of intracellular TJs contributes to the uncontrolled translocation of microorganisms, pathogens and antigens throughout the epithelium, resulting in an overactive mucosal immune response which subsequently leads to chronic inflammation, as in the case of inflammatory bowel disease (IBD) [5]. Crohn’s disease and ulcerative colitis, the two main disorders of IBD, represent chronic relapsing and remitting disorders of the gastrointestinal tract. IBD is a complex multifactorial disease driven by an inappropriate immune response of the mucosal immune system against endogenous microbes in humans with genetic predisposition [6]. The main focus of current IBD therapy is on anti-inflammatory agents and intestinal immune-modulating therapies to control inflammation in order to maintain barrier integrity and ultimately achieve mucosal healing [7]. Corticosteroids, 5-aminosalicylates, thiopurines and anti-TNF are examples of different types of available IBD therapies [8]. However, the side effects of these therapeutic agents and their failure to maintain the remission period in all IBD populations draw attention to urgently needed new treatment strategies [9]. As the altered composition and function of the intestinal microbiota in IBD patients plays a crucial role in the pathogenesis of IBD, the management of intestinal dysbiosis is one of the suggested future therapies [10,11].

The gastrointestinal tract covered with up to 10^14^ microorganisms is considered to be the most dense population in the world [12]. The number of bacterial species reported in this community is estimated to be in range of 500 to 1000 [13]. The intestinal microbiome is dominated by five phyla, *Bacteroidetes*, *Firmicutes*, *Actinobacteria*, *Proteobacteria*, and *Verrucomicrobia* characterized by 16S rRNA gene sequencing [14]. It is becoming increasingly clear that commensal bacteria and their metabolites are important mediators of the “cross talk” between the polarized epithelium and other types of mucosal cells, and that their interaction with the epithelium affects many aspects of gut barrier function [15].

Commensal bacteria play a vital role in gastrointestinal health [16]. They can exert their effect through various mechanisms, such as limiting the colonization by pathogens [17], the strengthening of barrier function [18], the production of butyrate as a source of energy for colonocytes [19,20,21], and the maintenance of immune system homeostasis [22,23]. Several studies have reported that the abundance of bacteria from the phyla *Firmicutes* and *Bacteroidetes* in the stool sample of IBD patients is significantly reduced, making them possible biomarkers for the diagnosis of many intestinal disorders [24,25,26]. The best known example is *Faecalibacterium prausnitzii*, the most abundant bacterial species in healthy individuals, and one of the main butyrate producers within the gut. This obligate anaerobe Gram-positive bacterial species belonging to the *Firmicutes* phylum has been associated with anti-inflammatory and epithelial barrier-strengthening properties as well as epithelial homeostasis [27]. Strikingly, the numbers of *F. prausnitzii* are significantly reduced in patients suffering from IBD [26]. Another important member of the *Firmicutes* phylum is the anaerobic Gram-positive species *Roseburia intestinalis*, which also hasbeen identified as a butyrate producer [28]. Only few in vitro and in vivo studies exist describing a potential immune- and epithelial barrier-modulation [29,30,31]. In contrast, the influence of *Bacteroides faecis*, an anaerobic Gram-negative species from the *Bacteroidetes* phylum, is completely unknown so far. Given the fact that *F. prausnitzii*, *R. intestinalis* and species from the *Bacteroidetes* phylum are underrepresented in the microbiota of IBD patients [32,33], a therapeutic supplementation in association with conventional therapies represents a promising perspective in the regulation and treatment of IBD [34,35,36,37]. However, for this purpose, it is imperative to initially elucidate the interaction between these commensal bacterial species and the cells of the gastrointestinal-system, prior to any protective effect studies in animals or humans. 

Intestinal microbiota plays a vital role in human health and disease, however, the underlying mechanisms of host–microbiota interactions and their impact on immune regulation remain unclear [38,39]. An in vitro simulation of the gastrointestinal tract can provide a useful insight into the behaviour of the intestinal microbiota [40].The host interaction with gut microbiota has been assessed through different in vitro models such as the exposure of intestinal epithelial cells to bacteria-free supernatants [41] or the direct co-culture-like Transwell system [42] microcarrier beads [43], human oxygen bacteria anaerobic (HoxBan) system [44], human gut-on-a-chip [45] and HuMix (human–microbial cross talk) microfluidic device [46]. Each of these human–microbial co-culture approaches has its benefits and drawbacks; the research questions and the parameters to analyse determine which in vitro system is best to be used.

Consequently, the main focus of this study was on the in vitro characterization of three commensal bacteria species, specifically selected for their potential protective properties against gastrointestinal inflammation. For this purpose, the interaction between live commensal bacteria, namely *F. prausnitzii*, *R. intestinalis* and to our knowledge, for the first time, *B. faecis*, on inflamed and non-inflamed human intestinal epithelial cells cultured on transwell inserts, was investigated, with a focus on bacterial adherence, cellular barrier integrity, anti-inflammatory potential and action on tight junction protein expression. 

## 2. Materials and Methods

### 2.1. Epithelial Cell Culture

Human colon carcinoma cell lines Caco-2 and HT29-MTX were purchased from Merck (Merck Co., Darmstadt, Germany). The cells were grown separately in tissue culture flasks in DMEM medium supplemented with 10% (*v*/*v*) FBS, 2mM L-glutamine and 1% (*v*/*v*) non-essential amino acids (MEM NEAA), and incubated in a humidified atmosphere (5% CO_2_, 37 °C). Both cell lines were subcultured up to a confluence of 80–90% and re-seeded into new culture flasks. For experimental studies, Caco-2 and HT29-MTX cells were seeded at a density of 0.75 × 10^5^ cells/cm^2^ on permeable polyester membrane transwell inserts with 0.4 μm pore size (Sarstedt AG & Co. Nümbrecht, Germany). The cells were cultured for 21 days to reach the full differentiation stage. The growth medium was refreshed every 2 days. The HT29–MTX and Caco-2 cells used in the experiments were from passages 12–30 and 10–33, respectively.

### 2.2. Bacterial Culture

*Faecalibacterium prausnitzii* strain A2-165 (DSM 17677), *Roseburia intestinalis* (DSM 14610) and *Bacteroides faecis* (DSM 24798) were tested in our set of experiments. All the bacterial strains were purchased from the Leibniz-Institute German Collection of Microorganism and Cell Cultures GmbH (Braunschweig, Germany). *F. prausnitzii* and *R. intestinalis* bacteria were routinely maintained at 37 °C in the brain-heart infusion medium supplemented with 0.5% (*w*/*v*) yeast extract (YBHI), 1mg/mL cellobiose, 1mg/mL maltose, 0.5 mg/mL L-cysteine and 5mg/mL hemin without agitation in an anaerobic atmosphere of 10% CO_2_, 10% H_2_, and 80% N_2_ (Mini-Max anaerobic work station Meintrup DWS Laborgeräte GmbH, Lähden—Holte, Germany).The same culture medium composition was used for the cultivation of *B. faecis*, however, without the addition of hemin. Bacterial growth was determined spectrophotometrically by monitoring the changes in the optical density at 600 nm (OD_600_). Growth curves were generated using the averages of three biological replicates with three technical replicates each. For all bacterial–cellular interaction experiments, the bacterial species were grown to the mid-logarithmic phase. The inoculums were centrifuged at (300× *g* for 10 min) at room temperature. The final bacterial pellet was collected and washed with PBS at pH7.4. The bacterial pellet was then re-suspended in sterile phosphate buffer (PBS) and adjusted to an OD of 0.5 at 600 nm which equals a bacterial concentration of 2 × 10^10^ colony forming units (CFU)/mL. The incubation of Caco-2 and HT29-MTX with the individual bacterial species and with a three species bacterial mix was performed in an anaerobic chamber at three different multiplicities of infections (100:1, 1000:1 and 10,000:1 bacteria/cell).

### 2.3. Bacterial Adherence to Intestinal Epithelial Cells

Caco-2 and HT29-MTX cells were seeded in 24-well plates (Greiner Bio-One; Cellstar, Frickenhausen, Germany) at a density of 0.75 × 10^5^ cell/well. The culture medium was changed every other day for 21 days. Bacterial strains were grown in the YBHI medium under an anaerobic condition at 37 °C and added to the cell monolayers at multiplicities of infection (MOIs) of 100:1 (6.4 × 10^9^ CFU/mL), 1000:1(6.4 × 10^10^ CFU/mL) and 10,000:1 (6.4 × 10^11^ CFU/mL). After 4 h, the cells were washed with PBS to removed non-adherent bacteria prior to trypsinization by 0.25% trypsin–EDTA solution (Gibco). Detached cells were lysed by cold distilled water and plated out in serial dilution steps on the YBHI agar plate. The number of viable bacteria was assessed by counting the CFU on agar plates incubated under an anaerobic atmosphere at 37 °C for 48 h. The adhesion was expressed as the percentage of the number of adhered bacteria to the total bacteria used for the experiment and calculated as: the percent adhesio*n* = P = *p**G/100 where P corresponded to the bacterial adherence and G corresponded to the control (CFU/mL is the initial and final count of bacteria)

### 2.4. Induction of Barrier Dysfunction

#### 2.4.1. Inflammatory Cocktail

Differentiated Caco-2 and HT29-MTX cell monolayers were treated basolaterally with a cocktail of inflammatory mediators (100 ng/mL TNF-α + 25 ng/mL IL-1β + 50ng/mL IFNγ and 10µg/mL lipopolysaccharide (LPS) for 10 and 48 h, respectively [8]. The cytomix + LPS stimulation mimics chronic intestinal barrier dysfunction associated with chronic inflammation. Cells without any treatment were used as negative control. 

#### 2.4.2. Bacterial Treatment

In order to investigate the therapeutic properties of *F. prausnitzii*, *R. intestinalis* and *B. faecis*, the inoculums of the commensals were prepared as described in Section 2.2. After cytomix + LPS stimulation, the cells were washed with PBS and incubated apically with the individual bacterial species and with a three species bacterial mix at MOI 1000:1 and 10,000:1 for 6 h.

### 2.5. Transepithelial Electrical Resistance (TEER)

The integrity of the Caco-2 or HT29-MTX cell monolayers was assessed using a Millicell-ERS volt-ohm meter (Merck Millipore corporate, Darmstadt, Germany). Cell monolayers were used for the experiments after 21 days post seeding. In every co-culture experiment, only the cell monolayers with TEER values exceeding 600 ohm/cm^2^ were chosen. This is a commonly accepted value, indicating the presence of fully differentiated cell monolayers. An insert without cells was used as a blank and to obtain the sample resistance, whilst the blank value was subtracted from all samples. The final unit area resistance was then calculated by multiplying the sample resistance values by the effective transepithelial capacitance (0.33 cm^2^). To check the effect of bacterial density on cell barrier permeability, differentiated Caco-2 and HT29-MTX cell monolayers were washed twice with PBS and incubated in anaerobic DMEM for 30 min prior to the incubation with all three bacterial species at three different multiplicities of infections (100:1, 1000:1 and 10.000:1 bacteria/cell) in an anaerobic chamber. The TEER values were measured and recorded before and at the selected time points (directly after bacterial inoculation, 3 and 6 h after exposure to bacteria). 

### 2.6. Determination of Paracellular Permeability 

Paracellular permeability was determined by the flux of the fluorescein isothiocyanate (FITC)-labeled dextran 4000 (FD-4) from Sigma Aldrich (St. Louis, MO, USA) through differentiated Caco-2 and HT29-MTX monolayers according to a method reported by Chelakkot et al., with minor modifications [47]. Briefly, Caco-2 and HT29-MTX cell monolayers were pre-incubated with the pro-inflammatory cytokine + LPS cocktail for 10 and 48 h, respectively, prior to bacterial treatment for 6 hours. Following the treatment, the monolayer was washed with PBS to remove any residual bacteria and the cells were pre-equilibrated for 1 h with HBSS buffered at pH 7.4 at 37 °C. Cells were carefully washed twice with PBS and 200 µL FD-4 solution (1 mg/mL in HBSS) was added to the apical side of the monolayers and incubated for 4 h. The samples (100 µL) were collected from the basal chamber under sink conditions at 0, and 240 min and fresh HBSS solution was substituted with the same quantity. The fluorescent intensity of these basolateral aliquots was determined after sampling using a multi-mode microplate reader (SpectraMax M3, Molecular Devices). The wavelengths of excitation and emission were 490 and 530 nm, respectively. All the tests were carried out as triplicate biological experiments.

### 2.7. Determination of Cytokine Secretion by ELISA 

The culture supernatants were collected from the basal chamber of cytokine-stimulated cells incubated w/o commensal bacteria, centrifuged (10 min 600× *g* at 4 °C) and then stored at −80 °C. Supernatants were analyzed for chemokine production according to the manufacturer’s protocol (Biolegend).

### 2.8. Immunofluorescence Staining

Fully differentiated Caco-2 and HT29-MTX cell monolayers were stimulated with the pro-inflammatory cytokine/LPS cocktail. Subsequently, the cells were treated with bacteria individually and in combination for 6 h as described above. Following these treatments, the monolayers of both cell lines were washed with PBS and fixed with 75% (*v*/*v*) ethanol in PBS for 30 min. Fixed monolayers were permeabilized with 75% (*v*/*v*) acetone in PBS for 3 min at −20 °C. Prior to the subsequent staining and detection, the monolayers were washed and blocked for 40 min with 1% (*w*/*v*) bovine serum albumin (BSA) in PBS. The cells were then incubated with either mouse anti-occludin (sc-133256, 1:150 Santa Cruz Biotechnology) or rabbit anti-claudin-2 (ab53032, 1:200, Abcam) primary antibodies overnight at 4 °C. The cells were rinsed again with washing buffer (PBS + 0.05% Tween-20) three times every five minutes, followed by incubation with the Alexa Fluor 488 goat anti-mouse secondary antibody, (Invitrogen/ThermoFisher, 1:500) and Alexa Fluor 488 goat anti-rabbit secondary antibody (Invitrogen/ThermoFisher, 1:500) for 1 h at room temperature. The cells were rinsed again with washing buffer, and finally incubated with 1µg/mL Hoechst 33,342 from Sigma Aldrich (St. Louis, MO, USA) for 10 min to stain all the nuclei. The membranes were separated from the Transwell insert using a scalpel. The permeable support membrane was then mounted cell side up between a slide and coverslip with a mounting medium. The microscopy of the mounted membranes was performed on a Zeiss LSM 510 Meta Confocal Laser Scanning Microscope (Carl Zeiss AG, Oberkochen, Germany).

### 2.9. Western Blot Analysis

Cell monolayers were grown on 24-mm Transwell inserts at a density of 2 × 10^5^ cells/well and cultured for 21 days. Caco-2 and HT29-MTX cells monolayers were treated with the pro-inflammatory cytokine + LPS cocktail and bacteria (MOI 1000:1) as described before. Cells were lysed with lysis buffer containing phenylmethylsulfonyl fluoride (PMSF), sodium orthovanadate and a phosphatase inhibitor cocktail (Santa Cruz Biotechnology, Santa Cruz, CA, USA). After 30 min incubation in ice, the cell lysates were centrifuged at 12,000 *g* for 30 min at 4 °C. The supernatant was removed and the protein concentration was measured employing a Pierce™ BCA protein assay kit (Thermo Fisher scientific, Waltham, MA, USA). Approximately 25 µg of each sample was loaded per well on polyacrylamide-separating gels (12% for claudin-2 and occludin). Protein was transferred to nitrocellulose membranes using a Bio-Rad Semidry transfer apparatus. Transfers were conducted at 15 V for 30 min. Membranes were subsequently washed with tris-buffered saline/1% Tween-20 (TBS-T) and blocked in 5% BSA for 1 h following incubation with primary antibody (polyclonal anti—claudin-2 and monoclonal anti-occludin ), diluted 1:1.000 in 5% BSA at 4 °C overnight. The membranes were then incubated with IRDye^®^ 800CW Goat anti-Mouse and IRDye^®^ 800CW Goat anti-Rabbit IgG Secondary Antibody 1:10.000 dilutions in 5% BSA for 30 min at RT. After an additional round of washing, the bands were visualized and analyzed using an Odyssey imaging system (Li-Cor).

### 2.10. Statistical Analysis

Results are expressed as the means ± SD. For the TEER measurement, the change overtime in TEER values between the different groups were compared using the Mann–Whitney U test. One-way ANOVA was used for statistical analysis of ELISA and Western blot. A *p*-value of less than 0.05 was accepted as the level of statistical significance. All the analyses were performed with Graph-Pad Prism software (GraphPad Prism Inc., San Diego, CA, USA).

## 3. Results

### 3.1. Viability of Commensal Bacteria in Cell Culture Media

In order make our data more relevant for the in vivo situation, the Caco-2 and HT29-MTX cells were stimulated with live *F. prausnitzii*, *B. faecis* and *R. intestinalis* under anaerobic culture conditions. The ability of the commensal species to survive in an anaerobic cell culture medium (DMEM) was determined by measuring the culture optical density (OD600nm) over 12 h. After 6 h of incubation in the cell culture medium, all three bacterial species showed a decrease in the OD600nm compared to the previous time points. In contrast, the incubation in the supplemented YBHI medium showed no decrease in OD600 (Appendix A). Based on these experimental results, a time frame of 6 h was chosen for the following in vitro experiments.

### 3.2. Bacterial Adherence to Intestinal Epithelial Cells

The adhesion of commensal bacteria to host cells is considered as an appropriate parameter to determine the colonization potential of a probiotic strain [48]. Therefore, in the present study, we quantified the adhesion capability of *F. prausnitzii*, *B. faecis* and *R. intestinalis* to the intestinal epithelial cells Caco-2 and HT29-MTX in anaerobic condition (Table 1). The differentiated epithelial cells were separately challenged with live *F. prausnitzii*, *B. faecis* and *R. intestinalis* using multiplicity of infection (MOI) values ranging from 100 to 10,000 in an anaerobic chamber. The number of adhering bacteria was dependent on the density of added bacteria to the cell lines. All the species tested in adherence to the Caco-2 and HT29-MTX cells at a low MOI of 100:1 only reached values below the detection limit of this assay. However, the maximum adhesion to both cell lines was shown at the MOI 1000:1. Among all three bacterial species, *B. faecis* showed the highest adhesion to the Caco-2 and HT29-MTX with values of 2.49 ± 1.14 and 6.40 ± 1.46 percent of the inoculum attached to the cells, respectively. *F. prausnitzii* also adhered to both the target cells; however, the capacity was not as pronounced compared to *B. faecis* (1.51 ± 0.65 and 1.32 ± 0.94%, respectively). Only very small numbers of *R. intestinalis* adhered to both cell lines, making it the least adhesive species among all three tested (0.11 ± 0.07% and 0.06 ± 0.01% to Caco-2 and HT29-MTX cell lines, respectively). Together, these results prove that all species are able to adhere to polarized epithelial cell layers in a species-specific manner. However, the MOI 1000:1 proved to be the most efficient ratio, as lower or higher bacterial numbers per cell did not further improve the results. 

### 3.3. Effects of Commensal Bacteria on Healthy Polarized Epithelial Monolayers

Then, we tested the influence of *F. prausnitzii*, *B. faecis* and *R. intestinalis* on intact and non-stressed or damaged Caco-2 and HT29-MTX-polarized monolayers. Changes in the TEER values after bacterial contact were used as a readout parameter for potential effects. We tested different MOIs (100:1, 1.000:1 and 10.000:1) for each bacterial strain at two different incubation time points (3 and 6 h). Cell monolayers without bacterial incubation were considered as a control group. TEER was immediately measured after the inoculation of Caco-2 and HT29-MTX with viable *F. prausnitzii*, *B. faecis* and *R. intestinalis*. As shown in Figure 1A–F, when differentiated Caco-2 and HT29-MTX cells were exposed to a low MOI of 100:1, the TEER values after 3 and 6 h of co-incubation did not differ significantly compared to the control. At a higher MOI of 1.000:1, the TEER value of Caco-2 cells incubated with *F. prausnitzii*, *B. faecis* and *R. intestinalis* showed a significant increase with the corresponding values of 24.7%, 14.3% and 18.3% after 6 h of incubation, respectively. Similar results were observed for HT29-MTX. Following co-incubation with *F. prausnitzii*, *B. faecis* and *R. intestinalis* for 3 h, an increase in the TEER was noted, which reached values of 29.4%, 23.4% and 10% compared to control in the 6 h incubation experiment, respectively. However, the TEER value induced by the *R. intestinalis* treatment did not reach statistical significance. Together, these experiments revealed a time-dependent influence of mainly *F. prausnitzii* and *B. faecis* on the TEER of polarized epithelial cells. The effects were more pronounced after 6 h of incubation. The MOI 1000:1 proved to be the most efficient ratio, as higher bacterial numbers per cell did not further improve the results.

### 3.4. Effects of Commensal Bacteria on Inflamed Intestinal Epithelial Cell Function

In the next set of experiments, we studied the effect of the three bacterial species on inflamed epithelial cell monolayers. The aim was to evaluate if the commensal species could support the healing and recovery of the stressed and damaged cell monolayers. Intact monolayers were first treated with a pro-inflammatory cytokine mixture and LPS and the damage was recorded via TEER measurements. Subsequently, *F. prausnitzii*, *B. faecis* and *R. intestinalis* were added at a MOI of 1.000:1 and the TEER values were again recorded 3 and 6 h after bacterial inoculation. As shown in Figure 2A,B, the untreated monolayer maintained its TEER level over the time course of the experiment in anaerobic chamber. However, in the treated monolayer, the cytomix + LPS stimulation dramatically reduced the resistance of Caco-2 cells by 48.75% after 48 h, and HT29-MTX cells by 56.25% after 10 h compared to untreated control. The cells stimulated with the the cytomix + LPS cocktail did not show an increase in the TEER value after changing to fresh medium without inflammatory cytokine. This strongly indicated that there was no spontaneous recovery of the monolayer TJ function. Further experimental results illustrated that both monolayer cell lines stimulated with cytomix + LPS showed a remarkable increase in the TEER value after 3 h of incubation with *F. prausnitzii*, *B. faecis* and *R. intestinalis* at MOIs of 1000 and 10,000. This effect was further sustained at 6 h post incubation. The treatment of cell monolayers with the bacterial three species mixture (MOI 1000:1) significantly restored the cytomix + LPS-induced barrier disruption at both time points (HT29-MTX 34.0 ± 24.3% and Caco2 35.0 ± 11.55%) compared to the cytomix + LPS-challenged epithelia in the absence of bacteria (Figure 2C). These results indicated that all three bacteria species, separately and in combination, could ameliorate the disruption of the epithelial barrier function caused by inflammatory cytokines and LPS in both Caco-2 and HT29-MTX cell lines. Of note, at the higher bacterial loads, almost the same protective effects were observed.

### 3.5. Testing the Paracellular Permeability of Epithelial Cell Monolayers 

In order to specifically address the changes in paracellular permeability, we employed FITC–dextran-4 (FD-4) diffusion, as this is a widely accepted indicator to monitor and quantify these processes. FD-4 flux was measured to evaluate the protective effect of the bacteria on epithelial monolayer integrity (Figure 3). In agreement with the TEER values, Caco-2 and HT29-MTX cell monolayers exposed to the pro-inflammatory cytokine + LPS cocktail showed a significant increase in FD-4 diffusion and flux (3- and 5-fold, respectively, *** *p* < 0.001) compared to the control, thus confirming severe barrier damage after pro-inflammatory treatment. Inflamed and damaged HT29-MTX cells showed a significant reduction in FT-4 flux after the single species incubation with *F. prausnitzii* (65.34 ± 22.35%), *B. faecis* (65.04 ± 2.61%) and *R. intestinalis* (55.79 ± 2.87%) (Figure 3A, ^#^
*p* < 0.05), compared to the cytomix + LPS stimulated cells in the absence of any bacterial treatment. The same trend was observed in the Caco-2 cell monolayers. The inflammatory cytokine and LPS mixture raised the paracellular FD-4 flux, and this increase in permeability was significantly restored by the treatment of the cells with *F. prausnitzii* at both 1000:1 and 10,000:1 MOIs (55.48 ± 23.51% and 42.41 ± 18.90%, ^#^
*p* ≤ 0.05). Furthermore, the *B. faecis* treatment of the inflamed and damaged Caco-2 cells resulted in a significant reduction in barrier permeability by (48.38 ± 24.81%) and (49.47 ± 13.90%), (Figure 3B, ^#^
*p* ≤ 0.05) for MOIs of 1000:1 and 10,000:1, respectively. Additionally, a significant decrease in paracellular permeability was found in the cells incubated with *R. intestinalis*, relative to a stimulated cell without bacterial incubation (MOI 1000:1 = 32.63 ± 11.40% and MOI 10.000:1 = 38.46 ± 12.40% ^#^
*p* ≤ 0.05). The co-administration of viable *F. prausnitzii*, *B. faecis* and *R. intestinalis* as a mixture significantly attenuated the increase in paracellular FITC–dextran transport due to the inflammatory cytokine/LPS cocktail treatment in both the Caco-2 (MOI 1.000:1 = 48.46 ± 29.10% and MOI 10.000:1 = 41.45 ± 27.10% ^#^
*p* ≤ 0.05) and HT29- MTX (MOI 1.000:1 = 38.40 ± 93.5.10% and MOI 10.000:1 = 32.27 ± 34.85% ^#^
*p* ≤ 0.05) cell monolayers, compared to the appropriate controls in the absence of any bacteria. Therefore, apart from the positive effects on TEER, these bacterial species have major beneficial recovery effects on paracellular leakage induced by inflammation.

### 3.6. Determination of Bacterial Effects on Host Cell Cytokine Secretion 

Then, in vitro experiments were carried out in order to assess whether the three commensal bacteria strains protect the cells by influencing the secretion of inflammatory mediators. After the co-culture of the cytomix + LPS stimulated Caco-2 and HT29-MTX cells with live *F. prausnitzii*, *B. faecis* and *R. intestinalis* (MOIs 1.000:1 and 10.000:1) for 6 h, the inflammatory chemokines IL-8 and MCP-1 were quantified in the supernatant of both cell lines by ELISA. As shown in Figure 4A–C, the stimulation of both cell lines with the cytomix + LPS cocktail resulted in a significant elevation in the production of IL-8 and MCP-1, compared to the non-stimulated control cultures (**** *p* ≤ 0.0001). However, the level of both chemokines in the basolateral media of Caco-2 monolayers treated with all three bacterial species decreased significantly at both MOIs (^####^
*p* ≤ 0.0001). Moreover, there were low levels of MCP-1 secretion in the HT29-MTX supernatants co-incubated with all three species, however, the values did not reach statistical significance for the treatment with *F. prausnitzii* (MOI 1.000:1 *p* < 0.0921 and MOI 10.000:1 *p* < 0.0592) and *B. faecis* (MOI 1.000:1 *p* < 0.0534). The IL-8 secretion level in HT29-MTX cells co-incubated with *F. prausnitzii*, *B. faecis* and *R. intestinalis* showed an approximately 3.5-fold reduction compared to cytomix + LPS stimulated cell monolayers (^####^
*p* ≤ 0.0001). Following the simultaneous treatment of both cell lines with three bacterial species, the detection levels of both IL-8 and MCP-1 decreased significantly. However, the reduction in HT29-MTX for MCP-1 was not statistically significant (*p* < 0.1437). In summary, the bacterial treatment of inflamed and damaged epithelial cell monolayers induced a rather anti-inflammatory cytokine response, thereby protecting the monolayers from further damage.

### 3.7. Immunofluorescence Microscopic Localization of Tight Junction Proteins

The expression and localization of the tight junction protein was assessed by immunofluorescence staining. The staining of occludin in the control sample of both epithelial cell lines demonstrated the localization at areas of cell contact and resembled a cobblestone pattern. The stimulation of cells by the pro-inflammatory cytokine/LPS cocktail altered occludin localization away from the cell surface. The faint immunofluorescence signal indicated tight junction disruption. However, the incubation of cytomix + LPS-stimulated monolayers with *F. prausnitzii*, *B. faecis* and *R. intestinalis* separately and collectively restored the occludin localization at areas of cell contact (Figure 5A,B). Furthermore, occludin was hardly detected in HT29-MTX cells, suggesting that its synthesis was affected. The stimulation of both epithelial cell lines with the cytomix + LPS cocktail increased claudin-2 expression unlike in control cells. However, claudin-2 in inflamed Caco-2 and HT29-MTX treated by all three bacteria species, separately and in combination, rescued the enhancement of claudin-2 expression caused by the inflammatory cytokines and LPS (Figure 5C,D). These results indicated that the intestinal epithelial tight junction function was impaired after the stimulation by the inflammatory cocktail, and that all three bacteria species, separately and in combination, could partly restore it.

### 3.8. Effects of Bacteria on Tight Junction Protein Expression in Epithelial Cell Monolayers

To investigate whether the changes in paracellular permeability on epithelial cell monolayers following the inflammatory cytokine/LPS cocktail and *F. prausnitzii*, *B. faecis* and *R. intestinalis* treatments were associated with modifications in TJ protein expression, the protein levels of the claudin-2 and occludin were quantified by Western Blot analysis. The stimulation of monolayers with inflammatory cytokines and LPS caused a significant decrease in occludin expression (Caco-2 0.55 ± 0.14-fold, *p* = 0.03 and HT29-MTX 0.58 ± 0.13-fold, *p* = 0.008) (Figure 6A–D) while the claudin-2 expression was markedly increased (Caco-2 1.32 ± 0.67-fold, *p* = 0.004 and HT29-MTX 0.75 ± 0.18-fold, *p* = 0.0009) (Figure 7A–D). This was concomitant with the drop in TEER and the increased paracellular permeability. A 6 h incubation of Caco-2 with viable *F. prausnitzii*, *B. faecis* and *R. intestinalis* attenuated the occludin level reduction by 1.60 ± 0.28, 1.91 ± 0.21-fold and 1.31 ± 0.44-fold, respectively (Figure 6A,B). Similar results were observed for HT29-MTX (Figure 6C,D). Following the co-incubation with *F. prausnitzii*, *B. faecis* and *R. intestinalis* for 6 h, an increase in occludin expression was noted with the corresponding value of 1.45 ± 0.20, 0.97 ± 0.41 and 0.61 ± 0.12-fold compared to the cytokine and LPS-treated group, respectively. However, the level of increase caused by *R. intestinalis* and *B. faecis* treatment did not reach statistical significance. The co-administration of the three bacterial species as a mixture led to a significant increase (*p* ≤ 0.05) in the level of occludin compared to the cytomix + LPS-treated group in both cell lines. 

The expression levels of claudin-2 protein in inflamed Caco-2 was downregulated after the treatment of cell monolayers with *F. prausnitzii* (0.50 ± 0.22-fold, *p* = 0.01), *B. faecis* (0.53 ± 0.35-fold, *p* = 0.007) and *R. intestinalis* (0.32 ± 0.22-fold, *p* = 0.1) alone and in combination (0.51 ± 0.16-fold, *p* = 0.009) compared to the cytomix + LPS-treated group without treatment (Figure 7A,B). A similar trend was observed for the claudin-2 expression level in HT29-MTX cells (Figure 7C,D). The co-incubation of the inflamed cell with *F. prausnitzii*, *B. faecis* and *R. intestinalis*, individually and in combination, significantly reduced the claudin-2 expression by 0.39 ± 0.32-fold, 0.44 ± 0.09-fold, 0.41 ± 0.09-fold and 0.48 ± 0.01-fold, respectively.

## 4. Discussion

The adhesion capability of bacteria to intestinal epithelial cells promotes the residence time and affects the ability of probiotic strains to regulate the immune response. However, little is known, as to how the adhesion ability is correlated with immunomodulation [49]; there are indications that the adherence of some probiotic bacteria to the gastrointestinal mucosa might be essential to stimulate the host’s immune system [50]. Therefore, this property of bacteria is considered critical, affecting the host intestinal health. Many studies have used Caco-2 and mucus-secreting HT29-MTX cell lines to assess the adhesion ability of putative probiotic strains [51]. *Lactobacilli* and *Bifidobacteria* are the most common probiotics, and their adhesion properties to intestinal epithelial cells have been studied extensively [52,53]. However, to date, there has been only one study to investigate the adhesion capacity of *F*. *prausnitzii* [54] and to the best of our knowledge, there has been no study to evaluate the adhesion property of *B. faecis* and *R. intestinalis* to intestinal epithelial cells. In the present study, the Caco-2 and HT29-MTX cell lines were selected to determine the adhesion behavior of three commensal species. All species showed adhesion to the used cell lines, however, the adhesion level of *B. faecis* and *R. intestinalis* were greater to HT29-MTX cell line. These data corresponded with the finding by Schillinger et al., 2005, who showed that the adhesion of all probiotic strains varies among strains, though high adhesion rates to HT29-MTX compared to other cell lines were observed [55]. Gopal et al., 2001 used HT29, Caco-2 and HT29-MTX cells to assess the adherence ability of two *L. acidophilus* and two *L. rhamnosus* strains and reported the higher affinity of bacteria to adhere to the latter cell line. Moreover, they hypothesized that this behavior might be due the physical entrapment of bacteria in the mucus layer rather than the higher association of this cell line for strains [56]. 

In our study, *F. prausnitzii* showed almost the same percentage of adherence to the Caco-2 and HT29-MTX cell lines with per cent adhesion values of 1.51 ± 0.65 and 1.32 ± 0.94, respectively. Whereas Martin et al., 2017, reported no adherence of *F. prausnitzii* to HT29 cell lines, but indicated an adhesion level of up to 20% (relative to *Lactobacillus rhamnosus* GG) after the addition of mucin to HT29 cell line [54]. Adhesion in the gut is a complex process, affected by several factors such as nonspecific and specific interactions, which include hydrophobic interactions, cation-bridging and receptor–ligand binding [57]. The exact mechanism in which the three commensal bacteria adhered to IECs is yet to be investigated. 

Numerous in vivo studies have shown that the composition and diversity of the intestinal microbiota in IBD patients is strongly altered [58]. Furthermore, the depletion of specific species of the commensal bacterial lineages, such as *F. prausnitzii* and *R. intestinalis* was previously demonstrated. *F. prausnitzii* is a commensal species and currently in the focus of research. This species represents more than 3.5% of the total mammalian intestinal population and is one of the most abundant butyrate-producing bacterial species with key-importance in host health [59]. In only a few studies, similar beneficial effects have been described for the butyrate-producer *R. intestinalis*. For *B. faecis* from the *Bacteroidetes* phylum no information is available so far. 

Prior to any in vivo animal experimental study, it is mandatory to gather functional information using cell culture systems. Consequently, we used differentiated Caco-2 and HT29-MTX monolayers to simulate the intestinal epithelia barrier. By adding a pro-inflammatory cytokine and LPS stimulus to the differentiated monolayer we further simulated an inflammatory milieu similar to the situation in the GI-tract of IBD patients. We investigated the recovery of the general and specifically paracellular epithelial barrier function and immunomodulatory properties of the three anaerobic human commensal bacteria *F. prausnitzii*, *R. intestinalis* and for the first time, *B. faecis.* A maximum time frame of 6 h for this study was chosen based on the preliminary data (Appendix A) revealing that the bacterial growth rate decreased after 6 h incubation in cell culture medium. 

The stability of the intestinal barrier is of enormous importance to prevent the entrance of luminal substances and pathogens to the internal environment of the body [60]. Intestinal epithelial cells play a central role in separating the intestinal lumen from the underlying mucosal immune system [60]. The integrity of the single layer of epithelial cells is maintained by TJ proteins, such as occludin, claudins, junctional adhesion molecule and tricellulin [61,62,63,64]. The delocalization of these proteins is associated with barrier dysfunction, increased paracellular permeability [65], enhancing the entry of luminal antigens and provoking systemic inflammation, summarized as the leaky gut syndrome [66,67]. 

Our results showed that the pro-inflammatory treatment of IECs cells attenuated the expression of occludin, and markedly induced the expression of claudin-2. Furthermore, changes in TEER as marker for TJ integrity and the FITC–dextran flux were noted, whereas the intervention with *F. prausnitzii*, *R. intestinalis* and *B. faecis* indicated that all three species strengthened the cell-to-cell integrity. The commensal species restored the expression of both TJ proteins, which in turn improved the barrier function shown by TEER and FITC–dextran experiments as well as immunofluorescence staining. Such results could not be obtained to date for *B. faecis, s*o our results demonstrate for the first time the barrier-enhancing potential for this commensal species. Recently, Wang et al. reported that a 5-fluorouracil-induced TEER decrease in Caco-2 cells could be partially reduced by the supernatant of *F. prausnitzii* [68]. Furthermore, a few in vivo studies found that the application of *F. prausnitzii* or *R. intestinalis* in experimental murine colitis models improved the epithelial barrier, as shown by increased occludin expression in colonic tissue and lower FITC–dextran levels in serum samples [69,70]. Our results are consistent with the findings of previous studies which demonstrated that a high occludin expression and a decrease in the abundance of claudin-2 are associated with increased TEER values following treatment with probiotic strains [71,72,73,74,75]. These observations prompted us to hypothesize that increased occludin and decreased claudin-2 expression may contribute to the ability of the commensal species to strengthen the epithelial barrier. The influence on the expression of other TJ proteins not measured in this study may also be a possible mechanism of our commensal species as it is known from various probiotic strains that the enhancement of the intestinal barrier is mediated by the activation of several signaling pathways, resulting in a reorganization and increase in TJ proteins [76,77,78].

An imbalance between pro- and anti-inflammatory mediators produced by multiple immune and non-immune cells is a common feature in IBD [79]. Our data show for the first time that the secretion of IL-8 and MCP-1 by pro-inflammatory mixture-stimulated Caco-2 and HT29-MTX cells was inhibited by the treatment with all tested commensal species and reduced to the basal level. Our results are in agreement with other studies investigating the anti-inflammatory potential of commensal and probiotic bacteria. Another commensal species, namely *L. acidophilus* LA, was also able to downregulate the MCP-1 expression from TNF-α stimulated cells [80]. Furthermore, different strains of *Bifidobacteria*, *Lactobacillus* and *E. coli* could also reduce the TNF-α-induced IL-8 secretion [81,82,83,84]. Immunosuppressive effects have so far been shown for *F. prausnitzii* and *R. intestinalis*. Gene-expression analyses identified a lower expression of inflammatory mediators, such as NF-κB, IL-1 and TNF-α, whereas anti-inflammatory IL-10 was predicted to be activated after the *F. prausnitzii* stimulation of Caco-2 cells [85]. Furthermore, IL-8 production in IL-1β-stimulated Caco-2 respective TNFα stimulated HT29 cells could be reduced by *F. prausnitzii* supernatant [32,86]. *R. intestinalis* was able to increase the secretion of TGFβ in LPS-stimulated Caco-2 cells [87]. The in vivo application of live *F. prausnitzii* or the cell-free supernatant in experimental gut inflammation models attenuates the severity of inflammation and enhanced the intestinal epithelial barrier [32,68,86,88]. Similar effects have recently been demonstrated in a murine model of acute colitis for *R. intestinalis* [69]. The transcription factor NF-κB and the underlying pathway play an important role in the pro-inflammatory immune response and are activated by components of the pro-inflammatory mixture (IL-1β, LPS and TNF-α) [89,90]. The inhibition of the NF-κB signaling has a pivotal influence on downstream mechanism, like TNFα-induced barrier disruption, resulting in TEER enhancement and alteration in TJ expression [91,92]. It can be hypothesized that the observed commensal specific immunosuppression effect is mediated via NF-kB pathways, as this was already show for other enteric bacteria [93,94].

The beneficial and attenuating effects of the bacterial commensal species shown in our study could have different underlying mechanisms. First, bacterial adherence to their target cells, even in low numbers as shown here, could protect epithelial cells or even restore their normal function. Second, soluble secreted proteins, as most likely active in all previously published studies employing bacterial supernatants, could suppress NF-κB activity and signaling and therefore alter cytokine and/or chemokine profile. Third, protection effects could be promoted by bacterial surface proteins. All these potential mechanisms need to be studied in more detail in the future. Such effects could be recently demonstrated for the probiotic *Lactobacillus bulgaricus* in a *Helicobacter pylori in vitro* model, resulting in a decreased IL-8 expression [95]. 

To our knowledge, this study is the first investigating the influence of the commensal bacterial species *F. prausnitzii*, *R. intestinalis* and *B. faecis* as a single or mixed application on the intestinal epithelial barrier and inflammatory mediators in an inflammatory in vitro differentiated epithelial cell monolayer model. Since the integrity of the epithelial barrier is affected not only by inflammatory mediators, the evaluation of other parameters such as receptors, e.g., Toll-like receptors, transcription factors, e.g., NF-κB and signaling pathways, should be included to better understand the efficacy of *F. prausnitzii*, *R. intestinalis* and *B. faecis* on inflammation processes and intestinal permeability. The elucidation of the underlying mechanisms responsible for the effects in our study would further help to accelerate the process to clinical application.

## 5. Conclusions

To date, no report has shown the effectiveness of the mixture of the commensal species *F. prausnitzii*, *R. intestinalis* and *B. faecis* on healthy and inflamed human intestinal epithelial cells in vitro. All three strains, in single or mixed application, enhanced the barrier function by increasing TEER, decreasing the FITC–dextran flux, decreasing IL-8 and MCP-1 secretion and influencing TJ expression. The potential of enhancing the intestinal barrier and attenuation of inflammation in epithelial cells supports our hypothesis that the application of specific bacteria which are underrepresented in IBD patients could support common IBD therapy and thereby manage intestinal inflammatory disorders.

Consequently, our next study will transfer these in vitro results to an in vivo mouse model, which is currently underway. These data will help to identify and evaluate a possible probiotic activity of specific commensal species. Furthermore, the identification of the underlying mechanisms that contribute to the in vitro efficacy and other favoring effects with regard to probiotics will be the focus of further research. 

## Figures and Tables

**Figure 1 nutrients-12-02251-f001:**
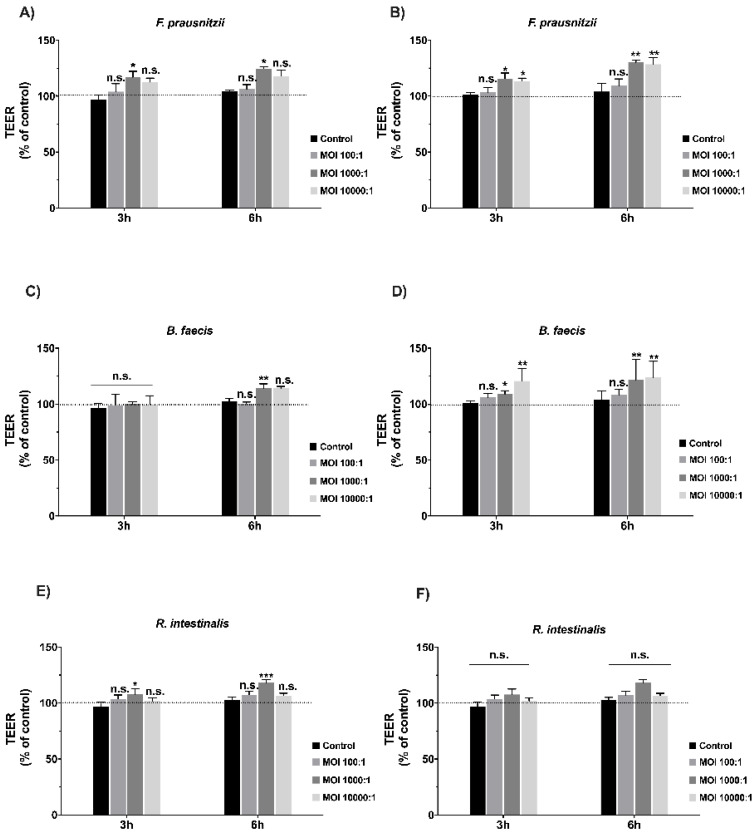
Effect of *F. prausnitzii*, *B. faecis* and *R. intestinalis* on the transepithelial electrical resistance (TEER) of epithelial cells. Caco-2 (**A**,**C**,**E**) and HT29-MTX (**B**,**D**,**F**) cell monolayers were incubated with all three bacterial species at three different multiplicities of infections (MOIs) in an anaerobic chamber. The transepithelial electrical resistance (TEER) was measured 3 and 6 h after the cells’ exposure to bacteria. Results are represented as the means ±SD (*n* ≥ 5). Data were analyzed with the Mann–Whitney U-test, * *p* ≤0.05, ** *p* ≤0.01, *** *p* ≤ 0.001; n.s., not statistically significant.

**Figure 2 nutrients-12-02251-f002:**
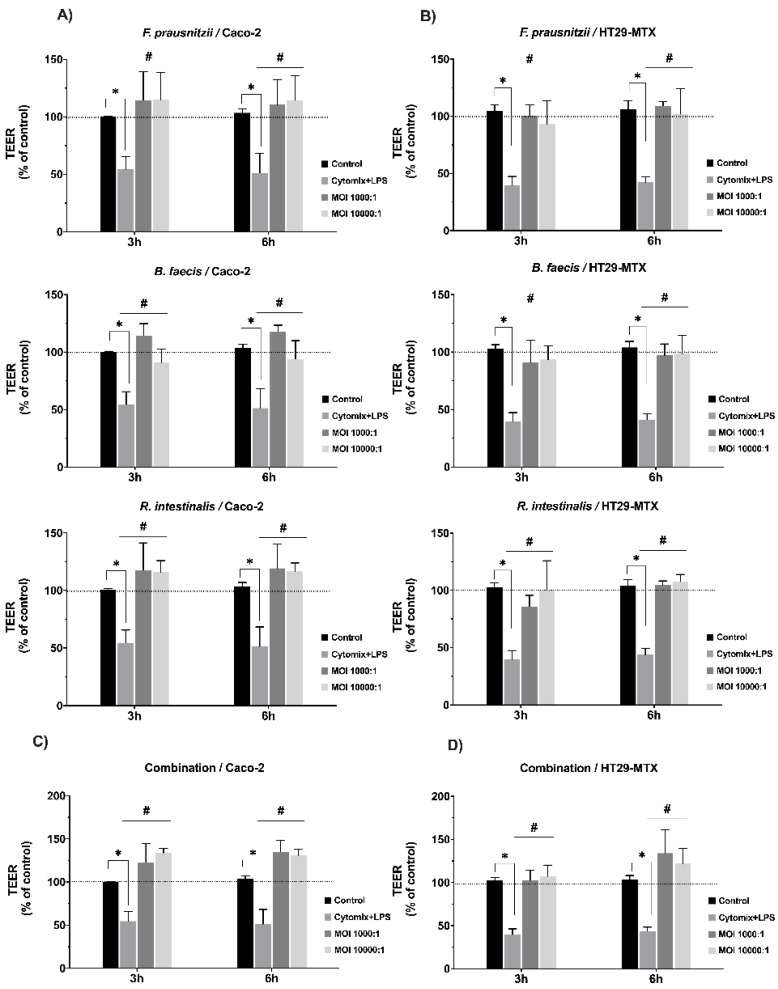
Effect of *F. prausnitzii*, *B. faecis* and *R. intestinalis* individually (**A**,**B**) and in combination (**C**,**D**) on the transepithelial electrical resistance (TEER) in Caco-2 and HT29-MTX cells monolayers. Both cell lines were treated basolaterally with cytomix + lipopolysaccharide (LPS) for 48 and 10 h, respectively. Cells without any treatment were used as a control. Data are presented as the means ± SD (*n* ≥ 5). * *p* ≤ 0.05) cytomix + LPS compared to the control and **^#^**
*p* ≤ 0.05 treated groups compared to cytomix + LPS, as determined by the Mann–Whitney U test.

**Figure 3 nutrients-12-02251-f003:**
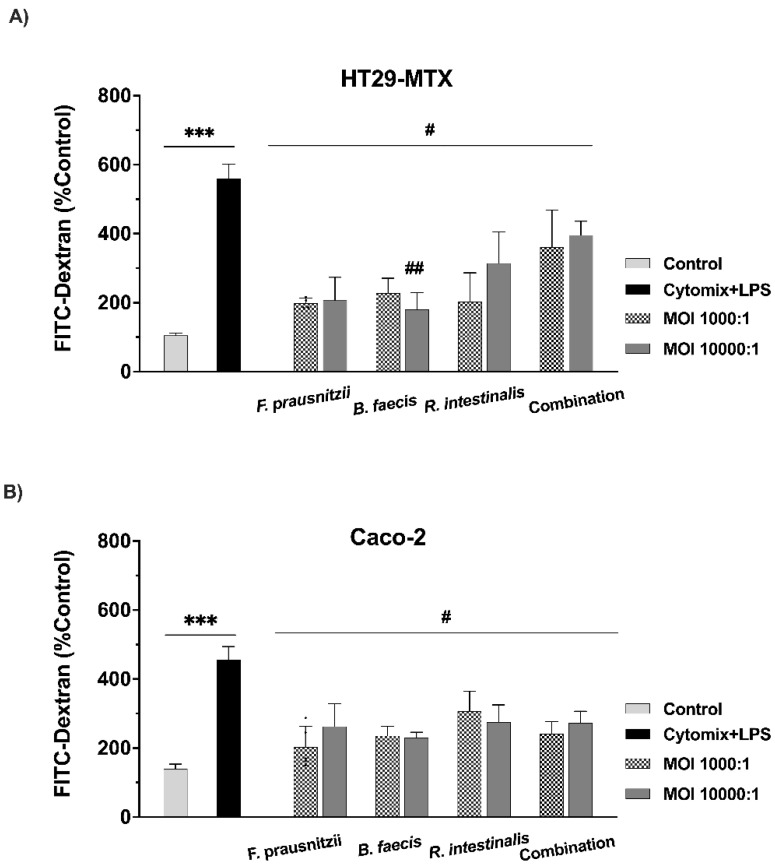
Paracellular permeability was determined by the flux of FITC–dextran through the differentiated HT29-MTX (**A**) and Caco-2 (**B**) monolayers. Cell monolayers were incubated with cytomix + LPS. After 6 h incubation with bacteria individually and in combination, the flux of 4-KDa FITC–dextran was measured. The control group received culture media. Data are presented as the means ± SD (*n* = 4). *** *p* ≤ 0.001 cytokine compared to control and ^#^
*p* ≤ 0.05, ^##^
*p* ≤ 0.01 treated groups compared to cytokine, as determined by the Mann–Whitney U test.

**Figure 4 nutrients-12-02251-f004:**
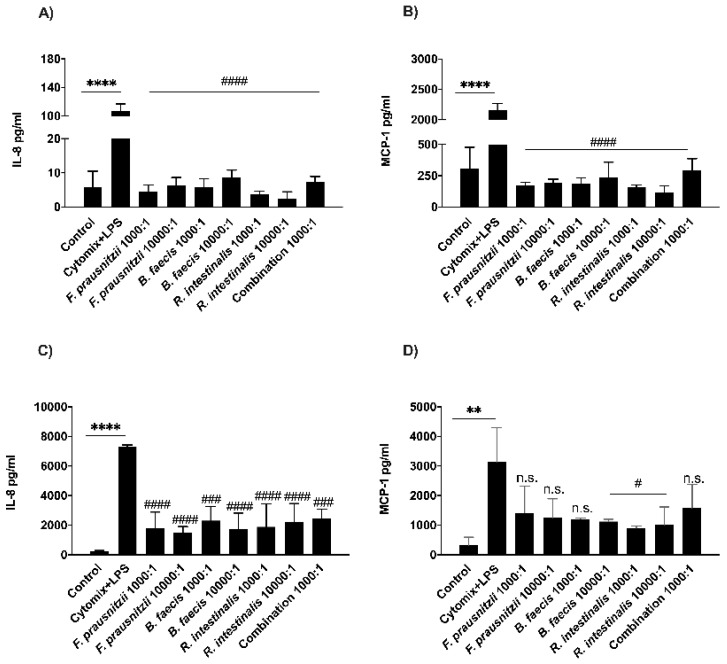
The secretion of IL-8 and MCP-1 in the inflamed Caco-2 and HT29-MTX monolayers. Epithelial cell monolayers were treated with cytomix + LPS following the incubation with bacteria individually or in combination for 6 h. Culture media sample was collected from the basal compartment. The concentration of IL-8 and MCP-1 secreted by Caco-2 (**A**,**B**) and HT29-MTX (**C**,**D**) cells were measured by multiplex flow cytometric bead array assay. ** *p* ≤ 0.01, **** *p* ≤ 0.0001 compared to the control, ^#^
*p* ≤ 0.05, *^###^ p* ≤ 0.001, *^####^ p* ≤ 0.0001, n.s., no statistical significance compared to cytomix + LPS, as determined by one-way ANOVA and presented as means ± SD (*n* = 3).

**Figure 5 nutrients-12-02251-f005:**
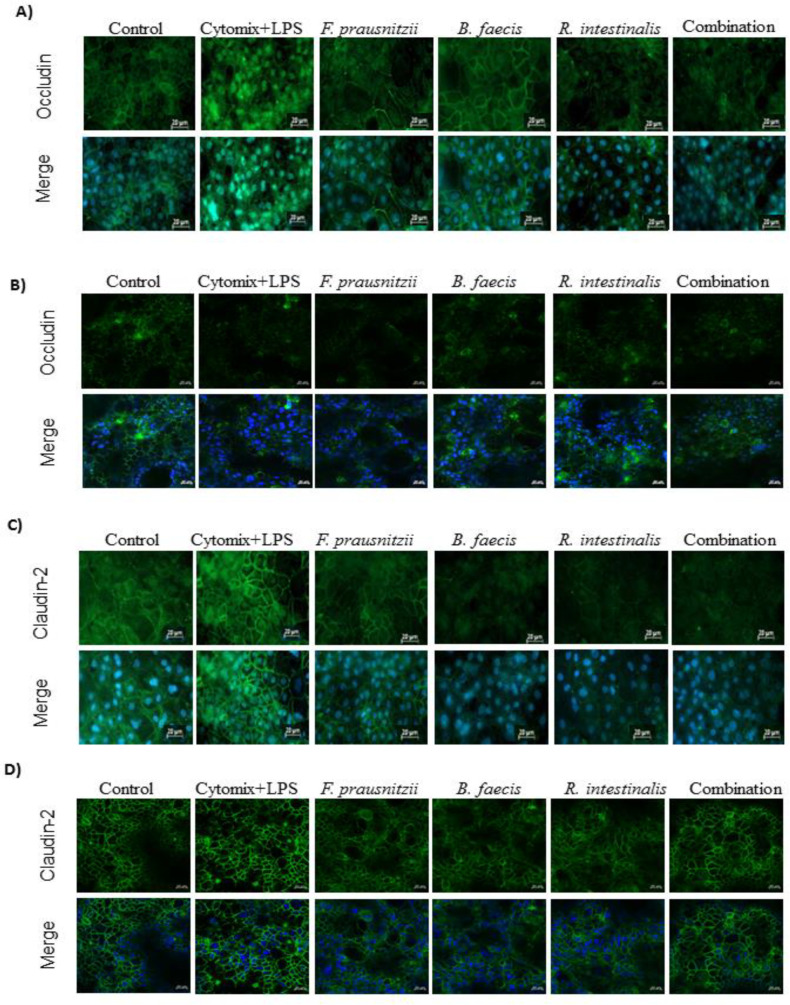
Effect of *F. prausnitzii*, *B. faecis* and *R. intestinalis* alone and in combination on the immunofluorescence localization of tight junction proteins. Both cell line monolayers were incubated with all three bacteria alone and in combination following incubation, cell monolayers were fixed and stained with anti-occludin and claudin-2 antibodies (green) and Hoechst (blue) and imaged by confocal microscopy. Immunofluorescence staining of Occludin in Caco-2 (**A**) and HT29-MTX (**B**) cell monolayers. Immunofluorescence staining of claudin-2 in Caco-2 (**C**) and HT29-MTX (**D**) cell monolayers. Images are of 40× magnification. Bar = 20 µm.

**Figure 6 nutrients-12-02251-f006:**
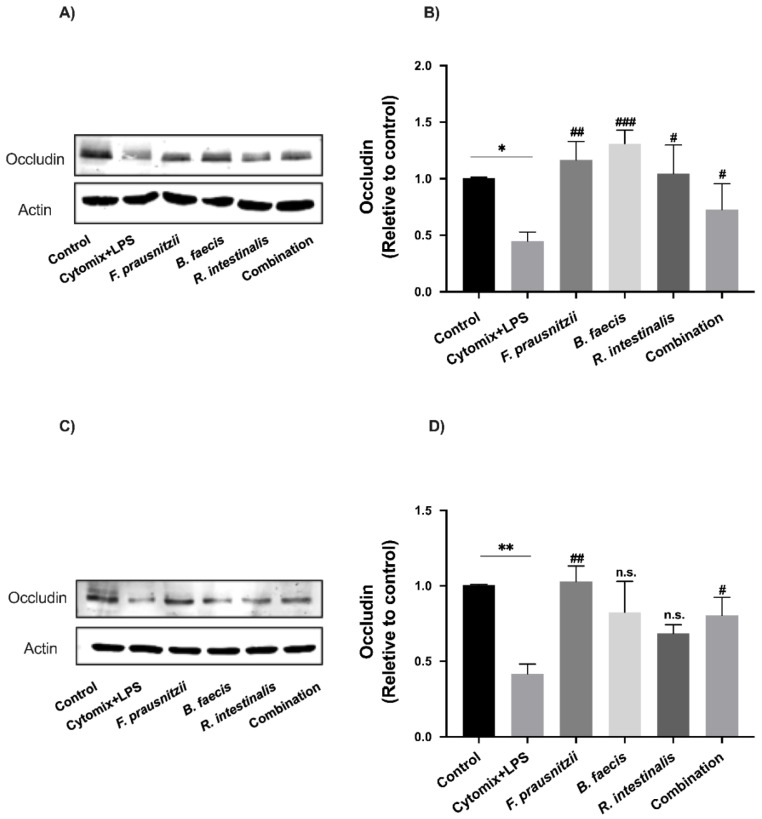
Effect of *F. prausnitzii*, *B. faecis* and *R. intestinalis* alone and in combination on occludin expression. Representative occludin immunoblot and relative densities of Caco-2 (**A**,**B**) and HT29-MTX cells (**C**,**D**). The protein bands were quantified through densitometry analysis and normalized to the intensity of the β-actin bands. Data were shown as the mean  ±  SD of three biological replicates (*n* = 3). * *p* ≤ 0.05 ** *p* ≤ 0.01 compared to control group. ^#^
*p* ≤ 0.05, ^##^
*p* ≤ 0.01, ^###^
*p* ≤ 0.001, n.s., no significant compared to the cytomix + LPS group as determined by one-way ANOVA.

**Figure 7 nutrients-12-02251-f007:**
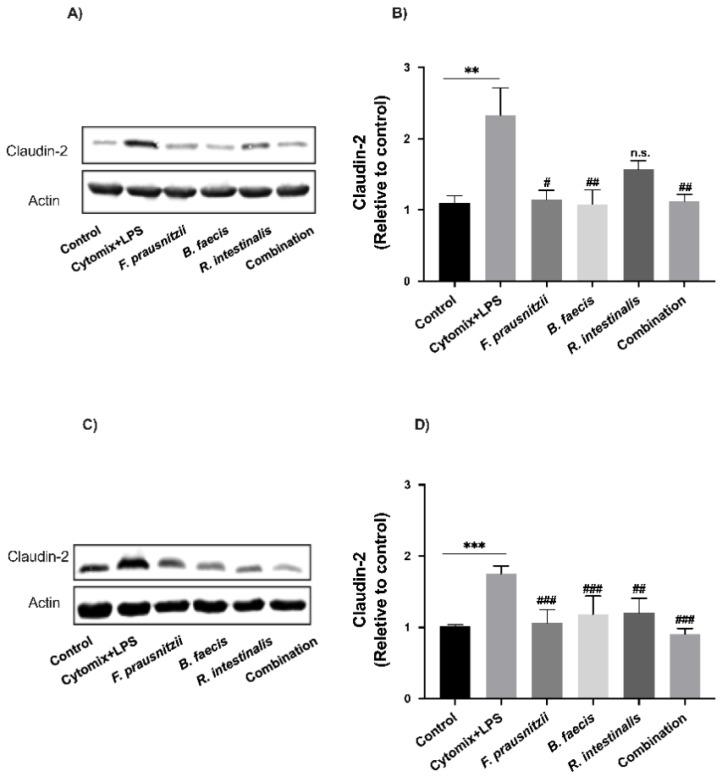
Effect of *F. prausnitzii*, *B. faecis* and *R. intestinalis* alone and in combination on claudin-2 expression. Representative claudin-2 immunoblot and the relative densities of Caco-2 (**A**,**B**) and HT29-MTX cells (**C**,**D**). The protein bands were quantified through densitometry analysis and normalized to the intensity of the β-actin bands. Data were shown as the mean  ±  SD of the three biological replicates (*n* = 3). ** *p* ≤ 0.01 *** *p* ≤ 0.001 compared to the control group. ^#^
*p* ≤ 0.05, ^##^
*p* ≤ 0.01, ^###^
*p* ≤ 0.001, n.s., no significant compared to the cytomix + LPS group as determined by one-way ANOVA.

**Table 1 nutrients-12-02251-t001:** Adhesion of *F. prausnitzii*, *B. faecis* and *R. intestinalis* to the differentiated intestinal epithelial cells Caco-2 and HT29-MTX at different multiplicities of infection (MOIs). The percent adhesion was calculated relative to the inoculums that were added to the cells for the adhesion assay. ND, not detected. Results are represented as the mean of triplicate experiments ± SD.

Bacterial Species	MOI	Relative Adhesion (% of Inoculum)	
		Caco-2	HT29-MTX
***F. prausnitzii***	100:1	ND	ND
***B. faecis***	100:1	ND	ND
***R. intestinalis***	100:1	ND	ND
***F. prausnitzii***	1000:1	1.51 ± 0.65	1.32 ± 0.94
***B. faecis***	1000:1	2.49 ± 1.14	6.40 ± 1.46
***R. intestinalis***	1000:1	0.11 ± 0.07	0.06 ± 0.01
***F. prausnitzii***	10,000:1	1.07 ± 0.50	0.02 ± 0.00
***B. faecis***	10,000:1	4.87 ± 2.69	5.39 ± 2.37
***R. intestinalis***	10,000:1	0.01 ± 0.00	0.01 ± 0.00

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
