# Peer review of "Barrier Protection and Recovery Effects of Gut Commensal Bacteria on Differentiated Intestinal Epithelial Cells In Vitro"

_nutrients, 2020, doi:10.3390/nu12082251_

Round 1
Reviewer 1 Report
The manuscript reported the effects of 3 commensal bacteria in the human gut on inflammation in vitro. It is an interesting connection to clarify bacteria-host interactions that would compile the available knowledge about the overall health. The manuscript is well-written and only a few minor issues that the authors need to address.
L55, L79, L441, L483, L484: Cite a reference
L220: Clarify when do you decide to choose one analysis than the other
L456: Delete “significantly”
L459: Add the exact percentage
L491: Define the direction of the alterations? Increased or decreased
L518: Gene not gen
L544: “in vitro” should be italic
Reviewer 2 Report
Review manuscript described by Mohebali et al. “Barrier Protection and Recovery Effects of Gut Commensal Bacteria on differentiated Intestinal Epithelial Cells in Vitro” is very interesting to me and significant especially in relation to host-microbial interaction in various inflammatory bowel diseases. There is some suggestions need to be addressed:
Major comments:
- The technique and methodology used for the co-culture of human intestinal cells Caco-2 and HT29-MTX cells with anaerobic bacteria is not appropriate to show host-microbial interaction. As, transwell inserts do not provide a vascular interface nor can they sustain luminal oxygen levels below 0.5%, which is required for co-culture of obligate anaerobes. Another major limitation is that in the medium in the dual-oxygen environment, the oxygenation level of the epithelial cells is not known, raising the question whether the cells are hypoxic. The methods like HoxBan system, organoid culture etc. could be implemented to prove these results.
- Culture of mammalian intestinal cells under hypoxic condition leads to activation of Hypoxia-Inducible Factor(HIF)-1 and its subsequent accumulation within the cells. Author need to address this issue with the current experiment in relation to host-microbial interaction.
Minor comments:
Scale bar for figure 5-C need to be provided.
